# Direct imaging of magnetohydrodynamic wave mode conversion near a 3D null point on the sun

Pankaj Kumar [1,2] ✉, Valery M. Nakariakov [3], Judith T. Karpen [2] & Kyung-Suk Cho[4,5]

Mutual conversion of various kinds of magnetohydrodynamic (MHD) waves can have profound impacts on wave propagation, energy transfer, and heating of the solar chromosphere and corona. Mode conversion occurs when an MHD wave travels through a region where the Alfvén and sound speeds are equal (e.g., a 3D magnetic null point). Here we report the direct extreme ultraviolet (EUV) imaging of mode conversion from a fast-mode to a slow-mode MHD wave near a 3D null point using Solar Dynamics Observatory/Atmospheric Imaging Assembly (SDO/AIA) observations. An incident fast EUV wavefront associated with an adjacent eruptive flare propagates laterally through a neighboring pseudostreamer. Shortly after the passage of the fast EUV wave through the null point, a slow-mode wave appears near the null that propagates upward along the open structures and simultaneously downward along the separatrix encompassing the fan loops of the pseudostreamer base. These observations suggest the existence of mode conversion near 3D nulls in the solar corona, as predicted by theory and MHD simulations. Moreover, we observe decaying transverse oscillations in both the open and closed structures of the pseudostreamer, along with quasiperiodic type III radio bursts indicative of repetitive episodes of electron acceleration.

Magnetohydrodynamic (MHD) wave mode conversion is a fundamental process that occurs in plasmas, where waves can change their mode (e.g., fast- to slow-mode or vice versa) as they propagate through the ambient medium. Mode conversion can occur when an MHD wave passes through a region where the plasma properties such as density or magnetic field strength change. In particular, effective coupling of slow and fast waves occurs in the regions where the sound and Alfvén speed are equal to each other. When a wave enters such a region, it can be partially reflected, transmitted, and converted into other wave modes, depending on the plasma properties and the angle of incidence of the wave [e.g.,[1]].

In the corona, regions of equal sound and Alfvén speeds could appear in magnetic null points (also called magnetic x-points or

neutral points). In 2D or 2.5D, null points are locations in the plasma where the strength of the magnetic field projected on a certain plane is zero, while the perpendicular component, i.e., the guide field, could remain non-zero. The latter point is important, as it prevents the Alfvén speed from going to zero. However, in 3D, there could be null points with all components of the field vector being zero [e.g.,[2]]. In the context of MHD-wave dynamics, a null point is a region where the fast magnetoacoustic speed has a minimum, making a null point a fast magnetoacoustic cavity or resonator. A fast wave that propagates in the vicinity of a null experiences refraction and turns toward the null point, leading to the effective accumulation of fast-wave energy in the vicinity of the null. The fast-wave front experiences a characteristic wrapping around the null point, well demonstrated by numerical

[1]Department of Physics, American University, Washington, DC 20016, USA. [2]Heliophysics Science Division, NASA Goddard Space Flight Center, Greenbelt, MD 20771, USA. [3]Centre for Fusion, Space and Astrophysics, Department of Physics, University of Warwick, Coventry CV4 7AL, UK. [4]Korea Astronomy and Space Science Institute, Daejeon 305-348, Korea. [5]University of Science and Technology, Daejeon 305-348, Korea. ✉e-mail: pankaj.kumar@nasa.gov

simulations of this process[3–5]. The accumulation of fast-wave energy near a null point leads to an increase in the wave amplitude, which can cause wave steepening and hence shocks and spikes in the electric current density[4,6]. In the finite-$\beta$ regime, the fast wave is converted into a slow wave that progresses outward from the null along the null-point separatrices.

The interaction between a fast mode and a null can trigger oscillatory reconnection[7,8], which could be responsible for the phenomenon of quasi-periodic pulsations (QPPs) during flares/eruptions[9–11], and could determine the duration and total amount of flare energy release. Furthermore, magnetic reconnection at a null point (also known as interchange or breakout reconnection) can explain the onset of a wide range of solar eruptive events, from small-scale jets to large-scale Coronal Mass Ejections (CMEs)[12–16]. These phenomena have been widely observed in solar bright points, pseudostreamers[17], and other null-point topologies[18–22]. Pseudostreamers (PSs) are particularly important because they play a significant role in the dynamics of the corona and formation of the solar wind[23], because they are embedded in open magnetic flux that expands into the heliosphere. In a broader context, the study of MHD mode conversion at 3D null points in the corona with the use of modern high-resolution imaging EUV telescopes opens up interesting perspectives for comprehending the transport of energy and the evolution of magnetic structures in solar, astrophysical, and laboratory plasmas.

Previous EUV observations and MHD simulations have inferred the appearance of a stationary wavefront when a fast-mode wave passes through a magnetic quasi-separatrix layer (QSL)[24,25]. This stationary or slowly-propagating wavefront is believed to appear due to a mode conversion process, where the fast-mode MHD wave is transformed into a slow-mode wave. Furthermore, quasi-periodic fast-mode EUV waves and associated radio bursts have been detected near the null point during eruptions[26]. Several MHD simulations have studied the behavior of MHD waves in the neighborhood of coronal null points[3,7,27]. To the best of our knowledge, direct imaging of theoretically predicted mode conversion at a 3D null point has not been reported before.

In this work, we present the direct observation of MHD wave mode conversion at a 3D null point in the solar corona. Using data from the Atmospheric Imaging Assembly (AIA) on board the Solar Dynamics Observatory (SDO), we have identified a clear signature of fast-to-slow mode conversion at a 3D null atop a pseudostreamer. Our analysis reveals the presence of a slow wavefront that is consistent with previous theoretical predictions and simulations. Furthermore, we observed a transverse oscillation (i.e., a kink oscillation) of both open and closed structures in the vicinity of the null point and indirect evidence for charged particle acceleration and the escape of electron beams, providing new insights into the physical conditions that lead to or are associated with mode conversion.

## Results
### Event overview and magnetic configuration
We analyzed SDO/AIA[28] images of a flare and associated eruption observed on May 9, 2014. The flare occurred behind the west limb in AIA, so the Geostationary Operational Environmental Satellites (GOES[29],) soft X-ray flux in 1–8 Å does not show any significant enhancement. The flare was associated with a fast halo CME (v=1100 km s$^{-1}$) that appeared in the Large Angle and Spectrometric COronagraph (LASCO[30],) C2 field of view at 02:48:05 UT. An SDO/AIA 171 Å image at 02:14:23 UT reveals the flare site and a nearby pseudostreamer (Fig. 1). A potential field extrapolation using the Helioseismic and Magnetic Imager (HMI[31],) magnetogram on May 5, 2014, illustrates a clear fan-spine/null-point topology (see Methods, subsection Magnetic Configuration). The width of the pseudostreamer dome is about 200″ and the height of the null is approximately 150″ (Fig. 1(b)). The Extreme Ultraviolet Imager [EUVI:[32,33]] on Solar TErrestrial RElations

Observatory Behind (STEREO-B) observed the same region near the east limb (Fig. 2). EUVI-B 171 Å images show the null-point topology from a different viewing angle (Fig. 2b) with a larger field of view than AIA. The flare began at about 02:14 UT (marked by the upper arrow in Fig. 2a). A filament in the flaring active region (lower white arrow in Fig. 2a) erupted during the flare. The EUVI-B 171 Å image at 04:14 UT (Fig. 2(b)) shows the flare arcade after the flare peak and filament eruption. EUVI-B 195 Å running-difference images ($\Delta t$=5 min) and the accompanying Supplementary Movie 1 reveal a fast EUV wave propagating northward from the eruption site (Fig. 2(c–e)). The EUV wave interacted with the pseudostreamer located north of the eruption site and continued to propagate toward the polar region.

### EUV wavefronts and kinematics
The AIA 193 and 211 Å channels provided the best view of the slow mode EUV wave that appeared near the null point. At 02:11:01 UT, the AIA 193 Å image shows the undisturbed null-point topology of the pseudostreamer (Fig. 3a). The fast EUV wave associated with a remote flare/eruption from the south passes through the null-point topology at about 02:20–02:24 UT (Fig. 3b, see accompanying Supplementary Movie 2). Following the passage of the fast EUV wave, a bright feature appeared near the null at around 02:24–02:25 UT (Fig. 3c) and disappeared shortly thereafter. This feature seems to be associated with the transverse displacement of the outer spine structure deflected by the propagating fast EUV wave. At around 02:26 UT, an EUV wavefront (S1) with an initial transverse length of about 150″ emerged near the null and propagated upward along the outer spine (Fig. 3d–g), observed in the AIA field of view until 02:34 UT. Moreover, during 02:29–02:33 UT, an EUV disturbance propagating downward (marked by a white arrow) along the fan was also detected (Fig. 3h, i).

The AIA 193 Å time-distance (TD) intensity plot (running difference, $\Delta t$=1 min) along slice SL1 (Fig. 4a, b) and (Supplementary Movie 3) shows the fast EUV wavefront (F) propagating south to north and passing through the pseudostreamer during 02:20–02:24 UT. The wavefront decelerates from 1430 to 970 km s$^{-1}$ (according to a second-order polynomial fit) within the 4 min interval. The bright features between 0 and 100 Mm in the TD plot during 02:23–02:50 UT are the filament material that moves radially upward slowly after the passage of the fast EUV wavefront. The TD running-difference intensity plot along SL2 (Fig. 4c, d and Supplementary Movie 4) shows slow fronts (denoted by S1, S2) propagating along the outer spine during 02:26–02:40 UT. Furthermore, we observed a downward-moving EUV disturbance (S3) along the fan loops during 02:29–02:33 UT. Wavefront S1 was very bright and propagated upward at 110–270 km s$^{-1}$ in the AIA field of view. Wavefront S2 was faint, propagated with approximately 150 km s$^{-1}$, and disappeared from the AIA field of view. Wavefront S3 propagated downward with about 216 km s$^{-1}$ along the fan loops simultaneously with S1.

### Radio bursts
Type II radio bursts are characterized by relatively slow frequency drifts within the dynamic radio spectrum and are believed to originate from plasma emission caused by shock-accelerated electrons in the solar corona[34]. In addition, type III radio bursts are signatures of high-speed electron beams injected along open field lines during the magnetic reconnection process[35]. Radio Solar Telescope Network (RSTN) radio flux density at 245 MHz (1-s cadence) reveals quasi-periodic radio bursts during 02:20–02:24 UT and a little enhancement during 02:27–02:28 UT (Fig. 5a). On the other hand, the 410 MHz flux profile (red) shows a single burst during 02:21–02:22 UT. RSTN high-frequency data (610–15400 MHz) did not show any microwave emission during the eruption. The Learmonth observatory dynamic spectrum (25–180 MHz) shows type III radio bursts from 02:21:30 UT to 02:23:30 UT, and a complex type II radio burst during 02:20–02:35 UT (Fig. 5b). The first peak of the 245-MHz radio burst (02:20–02:21 UT) is likely the extension of the type II burst. In addition, other quasi-

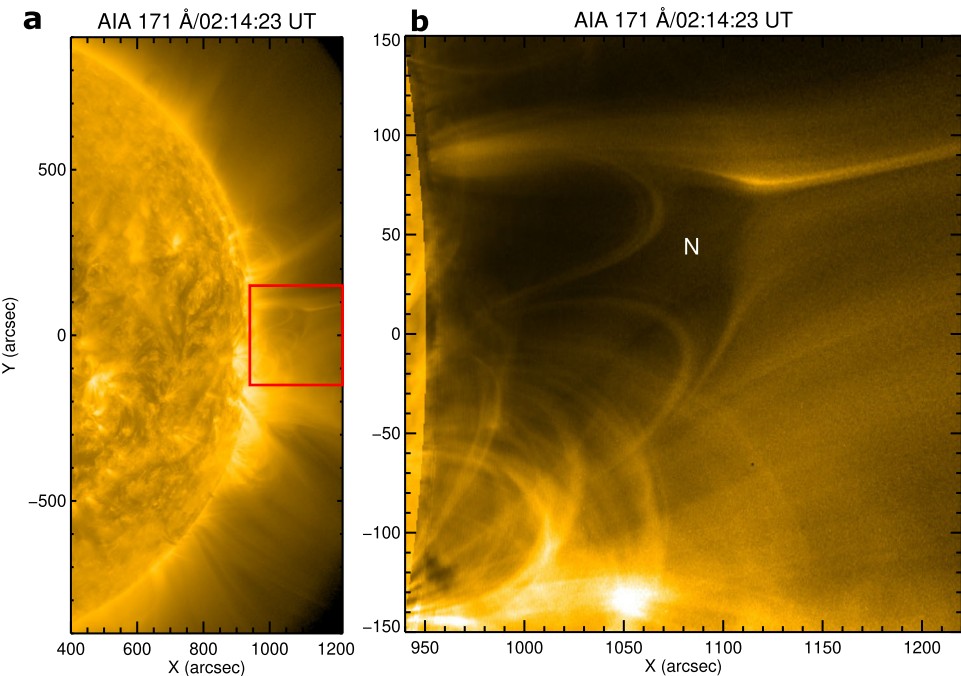

**Fig. 1 | Plasma configuration of the flare/eruption site. a** SDO/AIA 171 Å images of the eruption site showing a pseudostreamer (null-point topology inside the red box) on May 9, 2014. **b** A zoomed view of the pseudostreamer. The approximate position of the null point is marked by N.

periodic emissions during 02:21–02:24 UT match the type III radio bursts (Fig. 5a, b). The eruptive flare at the limb was associated with a coronal (observed by Learmonth observatory) and an interplanetary (2.6 kHz–16.025 MHz, WAVES instrument onboard Solar TErrestrial RElations Observatory Ahead (STEREO-A):[36,37]) type II burst (Fig. 5c, d), respectively indicating the presence of a low coronal and interplanetary shock. The low coronal type II burst began at about 02:20 UT and ended around 02:45 UT. It shows band-splitting (fundamental and harmonic) between 50–180 MHz. Using the Newkirk density model[38], the shock speed derived from the drift rate of the metric type II bursts was about 960 km s$^{-1}$. Type III radio bursts, a signature of electron beams injected along open field lines, began at 02:21 UT, in the WAVES-A data. At least four type III bursts were detected during a 40-min interval (2:20–3:00 UT).

## Transverse oscillations

A transverse oscillation (i.e., a kink or anti-symmetric transverse oscillation, see[39]) appeared shortly after the passage of the fast EUV wave through the PS. AIA 171 Å TD intensity plots along slices PQ and RS reveal four cycles of a decaying transverse oscillation in the open and closed structures of the PS during 02:22–02:46 UT (see Supplementary Movie 5, Fig. 6). The initial amplitude and period of the oscillation were about 5 Mm and 6 min. In addition, we identified quasi-periodic type III bursts at 245 MHz with a mean period of about 60 s, during the interaction of the fast EUV wave (02:21-02:24 UT) with the PS (Fig. 5a, b). Intriguingly, another small radio burst (about 02:27 UT) coincided with the emergence of the slow EUV wave (S1) near the null, while another type III at about 02:33 UT (Fig. 5c, d) coincided with the appearance of S2. The initiation times for the slow EUV waves S1 and S2 match closely with the first two cycles of the decaying transverse oscillation.

## Discussion

We analysed a multiwavelength EUV and radio observation of the interaction between an EUV wave and a coronal magnetic null point at the top of a pseudostreamer. The fast EUV wave accompanied a flare and a halo CME with a speed of 1100 km s$^{-1}$, which were initiated adjacent to the PS, nearly behind the limb. The fast EUV wavefront projected

on the plane of the sky decelerated from 1430 to 970 km s$^{-1}$ within a 4 min interval during its passage through the PS, at the same time as a metric type II radio burst. The speed derived from the drift rate of the type II radio burst, 960 km s$^{-1}$, is consistent with the measured speed of the EUV wave, suggesting that the EUV wave was a fast magnetoacoustic shock wave. We conclude that the type II radio burst was generated when the shock passed through the high-density PS and accelerated ambient electrons. Because coronal structures such as the PS are regions of low Alfvén speed and high density, intensification of the shock there would yield enhanced energization of a large population of electrons and the associated type II radio burst[4,6,40–42].

We detected in the AIA 171, 211, and 193 Å channels a slowly propagating EUV intensity disturbance (S1) near the null point shortly after the passage of the EUV wave. The wavefront propagated at the projected speed of 110–270 km s$^{-1}$ along the outer spine above the null. This behavior is quite consistent with theoretical modeling of a fast-wave interaction with a null point [e.g.,[3,4]]. The typical coronal sound speed ($c_s = 152\sqrt{T\,(\mathrm{MK})}$, where $c_s$ is the sound speed and T is the plasma temperature) in a 0.7–2.0 MK plasma is 120–215 km s$^{-1}$, which is comparable to the phase speed of the slow wave. We observed acceleration of the slow wave along the magnetically open stalk of the PS, which may be attributed to changes in the local temperature or to the change of the angle between the line of sight and the local wave vector. Therefore, this propagating EUV disturbance is consistent with a slow magnetoacoustic wave behavior (see the discussion in Methods, subsection MHD simulation of mode conversion and Fig. 7).

According to the AIA 193 Å running-difference images, the wavefront appears wide and bright when it first emerges near the null point, but as it travels upwards along the open structures, it becomes narrow and faint. The second slow EUV wavefront (S2) appeared roughly six minutes after the initial wavefront (S1), propagating at about 150 km s$^{-1}$ along the same path as S1. Thus, S2 is also interpreted as a slow-mode wave. These observations provide the direct imaging of the fast magnetoacoustic mode conversion (fast-mode to slow-mode MHD wave) at a 3D null.

Shortly after the passage of the EUV shock, we observed in AIA 171 Å a transverse oscillation of the flank boundary of the PS structure

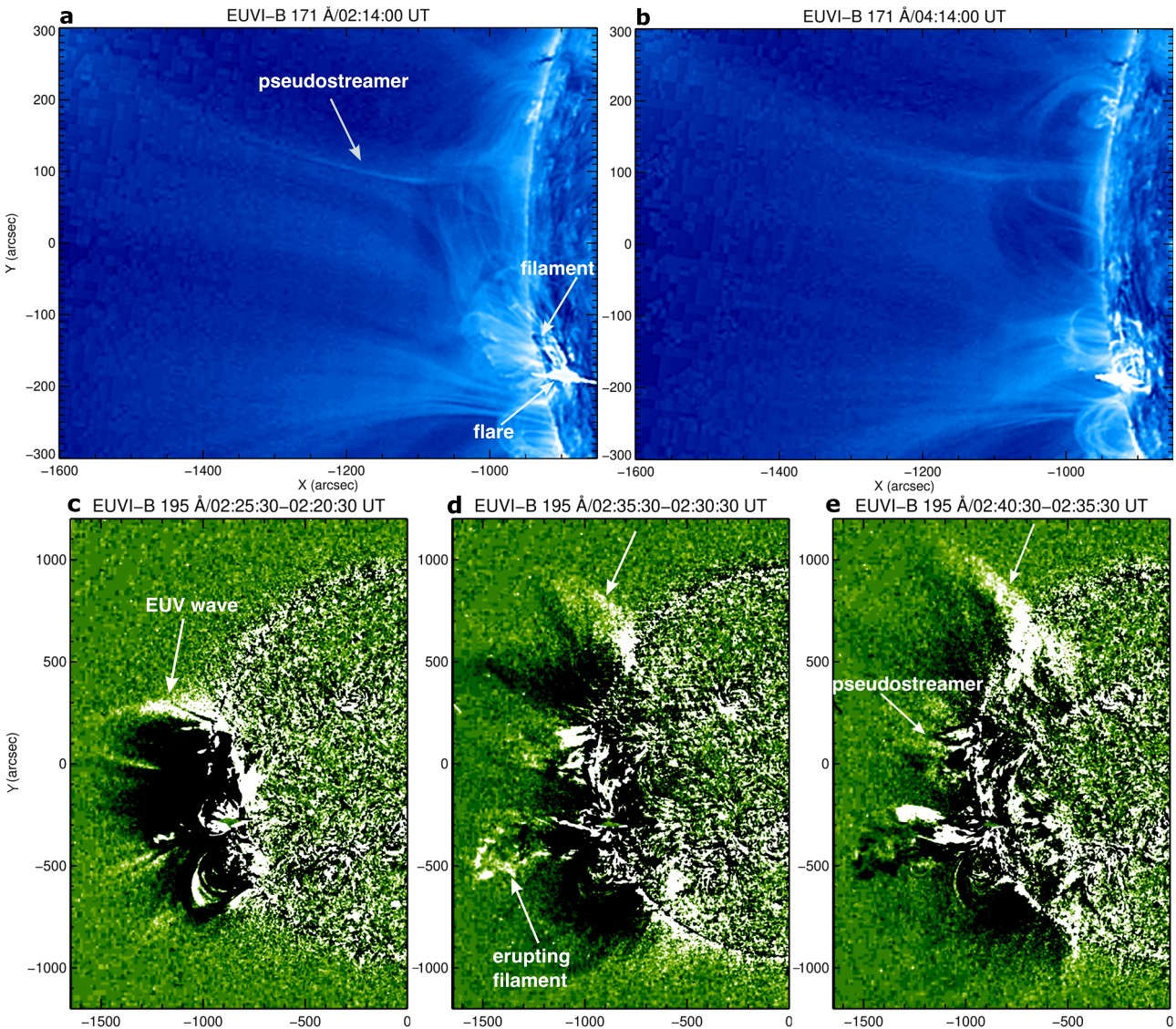

**Fig. 2 | STEREO EUVI-B images of the fast EUV wave associated with the eruptive flare and the adjacent pseudostreamer. a**, **b** 171 Å images showing the flare, filament, and pseudostreamer at the onset of the EUV wave and 2 h later on May 9, 2014. **c**, **d**, **e** 195 Å running-difference images of the eruption site and surroundings showing the propagating fast EUV wave. The upper white arrow marks the wave in all 3 panels. An animation of (**c**–**e**) is available as Supplementary Movie 1. The animation runs from 02:05:30 UT to 03:55:30 UT.

affecting both open and closed fields, with a period of about 6 minutes. The onset of slow disturbances S1 and S2 was clearly evident during the first two cycles (out of four) of the transverse oscillation, indicating a possible connection between the transverse oscillation and the onset of the slow-mode wave near the null. The connection between transverse oscillation and the appearance of slow-mode waves is speculative here. Both might be independent phenomena triggered by the incoming fast EUV wave[43], or one could be driving the other[4,27]. 3D MHD simulations should be conducted to establish whether there is a connection between these two phenomena.

Furthermore, the radio observations revealed the presence of type III radio bursts coinciding with the emergence of S1 (02:27 UT) and S2 (02:33 UT). These findings suggest that electron beams were injected into the open structures above the null. We speculate that the interaction of the fast wave with the null point induced quasi-periodic reconnection, which in turn released electron beams along the open field lines and generated type III bursts. This process possibly indicates reconnection driven repetitively by null-point oscillations, induced by an incident fast wave as shown in the numerical simulations of [7].

We performed a basic 2D numerical MHD simulation to demonstrate the behavior of a fast-mode wave at a null point (velocity shown in Fig. 7). This simulation focuses on the behavior of the fast-mode wave in a limited region around the null, without considering the large-scale magnetic configuration of the observed event. The incoming fast-mode wavefront undergoes complex deformation due to the magnetic topology near the null point. Slow-mode wavefronts appear after the interaction between the fast-mode wave and the null point and propagate outwards from the null along the separatrices. The observations are essentially consistent with the MHD simulation.

Published MHD simulations also offer evidence of mode conversion of MHD waves at PS null points[44–47]. For instance[46], showed a clear interaction of fast-mode waves with a PS null in a 2D MHD model. They detected slow-mode waves propagating upward along the PS stalk and simultaneously downward along the separatrix after the interaction, similar to the observations reported here. However, they found no evidence of oscillatory reconnection arising from the dynamics of the null itself, in contrast to other simulations [e.g.,[8]]. None of the above simulations found evidence for transverse oscillations of the PS open/

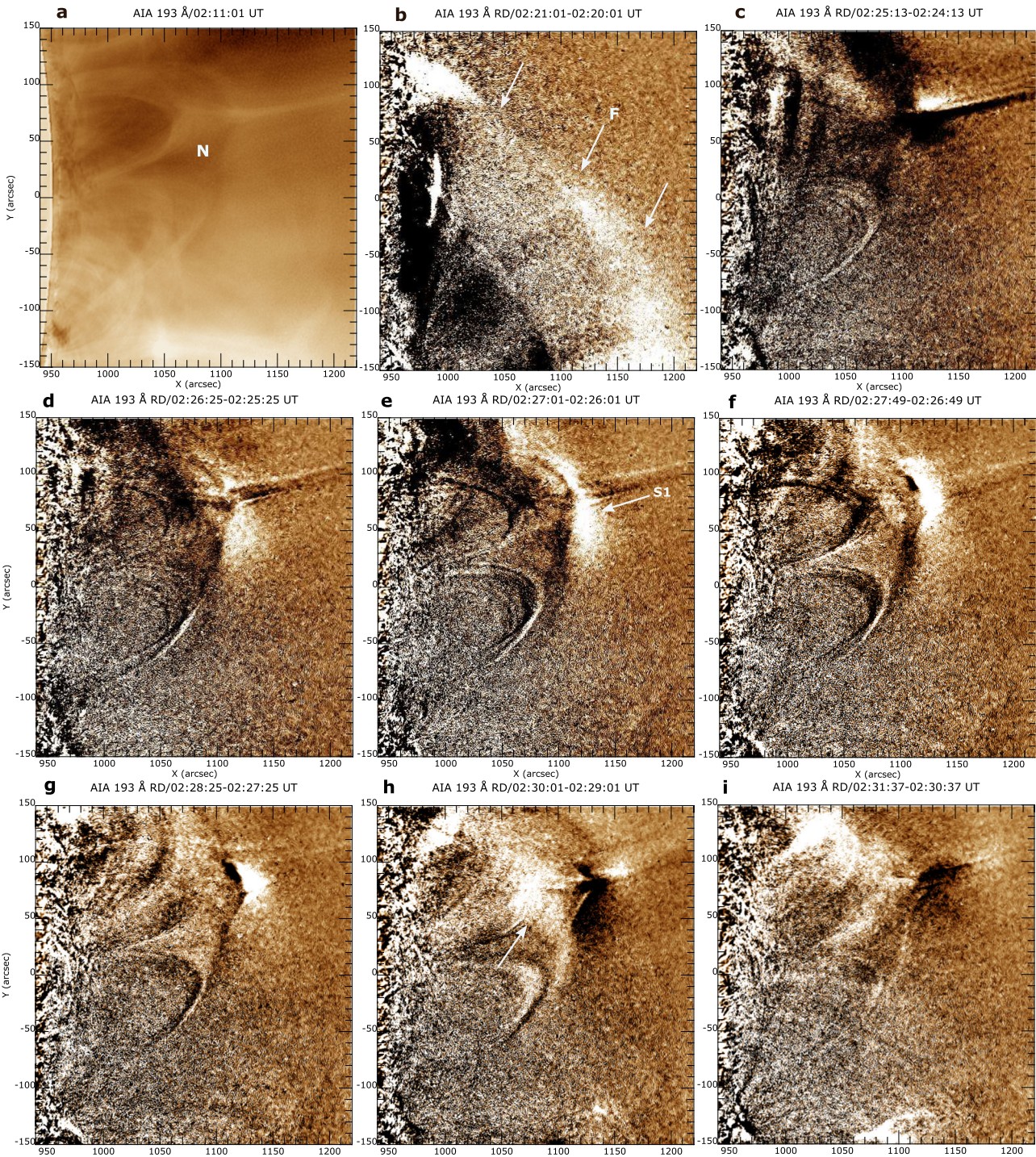

**Fig. 3 | Imaging of EUV wavefronts from the null. a–c** Sequence of SDO/AIA 193 Å images showing the fast EUV wave (F and white arrows) propagating through the null point N (May 9, 2014). **d–g** Appearance of slow mode EUV wave S1 near the null (marked by an arrow in (**e**)) and its propagation along the outer spine. **h, i** Downward moving disturbance (marked by the white arrow) along the fan. An animation of this Figure is available as Supplementary Movie 2. The animation runs from 02:11:49 UT to 03:06:25 UT.

closed structures. Other MHD simulations have investigated the propagation of Alfvén and magnetoacoustic waves in the vicinity of a 2.5/3D null point, examining the associated wave refraction and plasma heating[48,49].

Our observations suggest that the transformation of MHD waves at the null is a result of the mode conversion process, followed by indirect evidence of oscillatory reconnection. This insight may aid in explaining quasiperiodic intensity pulsations, particle acceleration in

jets, and recurrent jet outflows from the null. More realistic 3D MHD simulations would contribute to a better understanding of the interaction between MHD waves and null-point topologies, transverse oscillations, and the associated repetitive reconnection.

In conclusion, we studied the interaction of a fast-mode wave with a pseudostreamer. The observations reveal the direct imaging of the conversion of fast-to-slow mode MHD waves near a 3D null point in the solar corona. In addition, recurrent type III radio

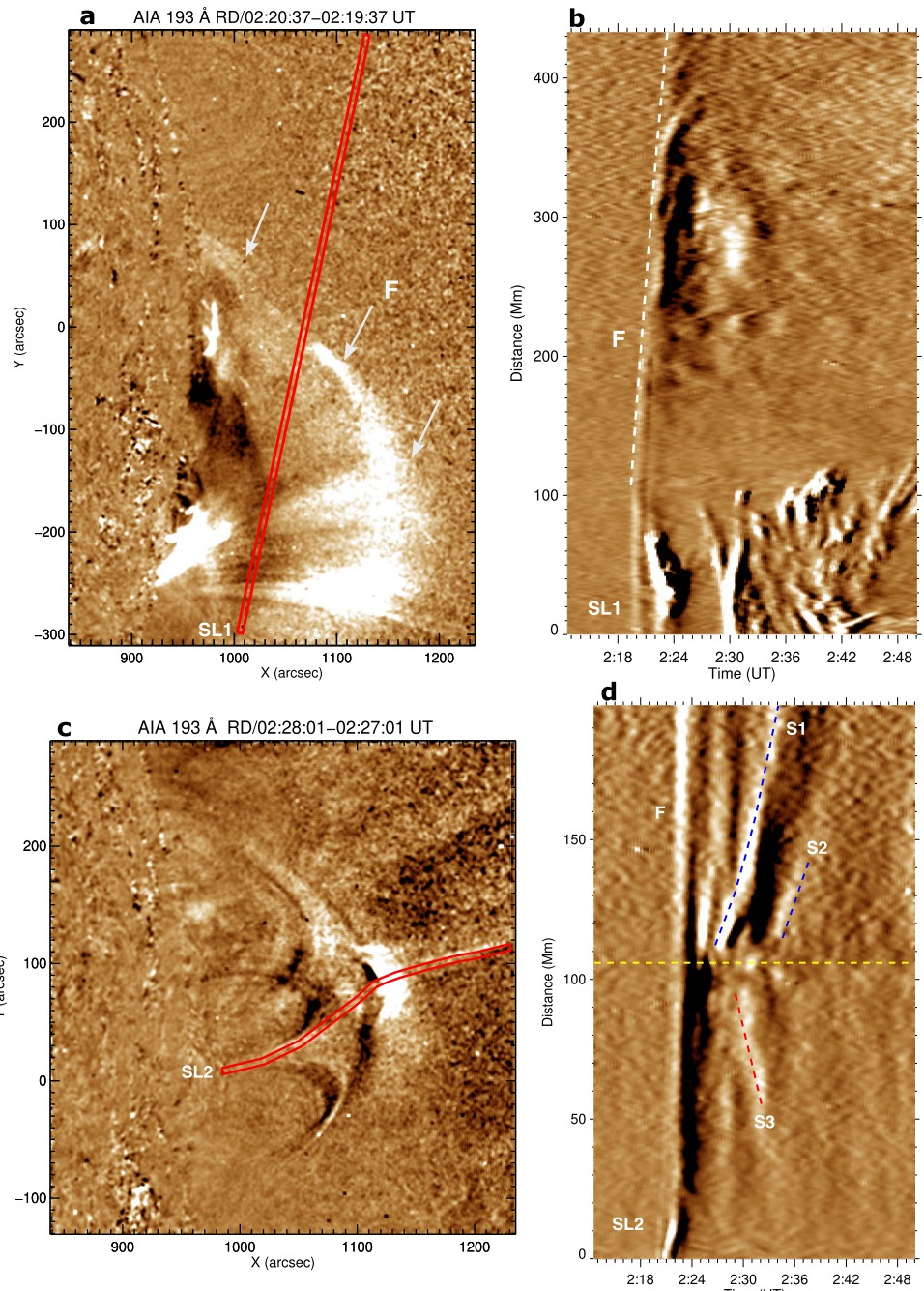

**Fig. 4 | Kinematics of the EUV wavefronts. a, c** SDO/AIA 193 Å running-difference images at selected times (May 9, 2014), showing the fast- and slow-mode waves and the slices SL1 and SL2 used to produce the accompanying time distance intensity plots. Distance is measured from the lower end of each slice. **b** Time-distance intensity (running-difference) plot along slice SL1 showing the fast EUV wavefront (marked by F). **d** Time-distance intensity (running-difference) plot along slice SL2 showing the propagating slow EUV wavefronts (marked by S1, S2, S3). The horizontal yellow dashed line indicates the approximate position of the null point. An animation of this Figure is available as Supplementary Movies 3, 4.

bursts suggest the excitation of oscillatory reconnection and associated particle acceleration. Given that the corona undoubtedly contains numerous null points, direct observation of mode conversion is important for understanding the role of MHD waves in energy transport and heating in the solar atmosphere. In addition, our study highlights the potential for mode conversion of EUV waves to supply energized seed particles for further acceleration by CME-driven shocks, a possibility that merits further investigation. Our results validate the complex process of mode conversion at the null point and provide strong motivation for future 3D modeling efforts in this field.

## Methods

### Data analysis

We analysed full-disk images of the Sun captured by the SDO/AIA, with a field-of-view of 1.3 $R_\odot$, a spatial resolution of 1.5" (0.6" pixel$^{-1}$, cadence=12s). We utilized AIA 171 Å (Fe IX, $T = 0.7$ MK), 193 Å (Fe XII, Fe XXIV, $T = 1.2$ MK and = 20 MK), and 211 Å (Fe XIV, $T = 2$ MK) images. The 3D noise-gating technique[50] was employed to remove noise from the AIA images. We also used the aia_rfilter routine (available in SSWIDL) to enhance the off-limb features in AIA images. We utilized SDO's HMI magnetogram to determine the magnetic configuration of the source region. STEREO EUVI-B observed this flare/eruption close to the east

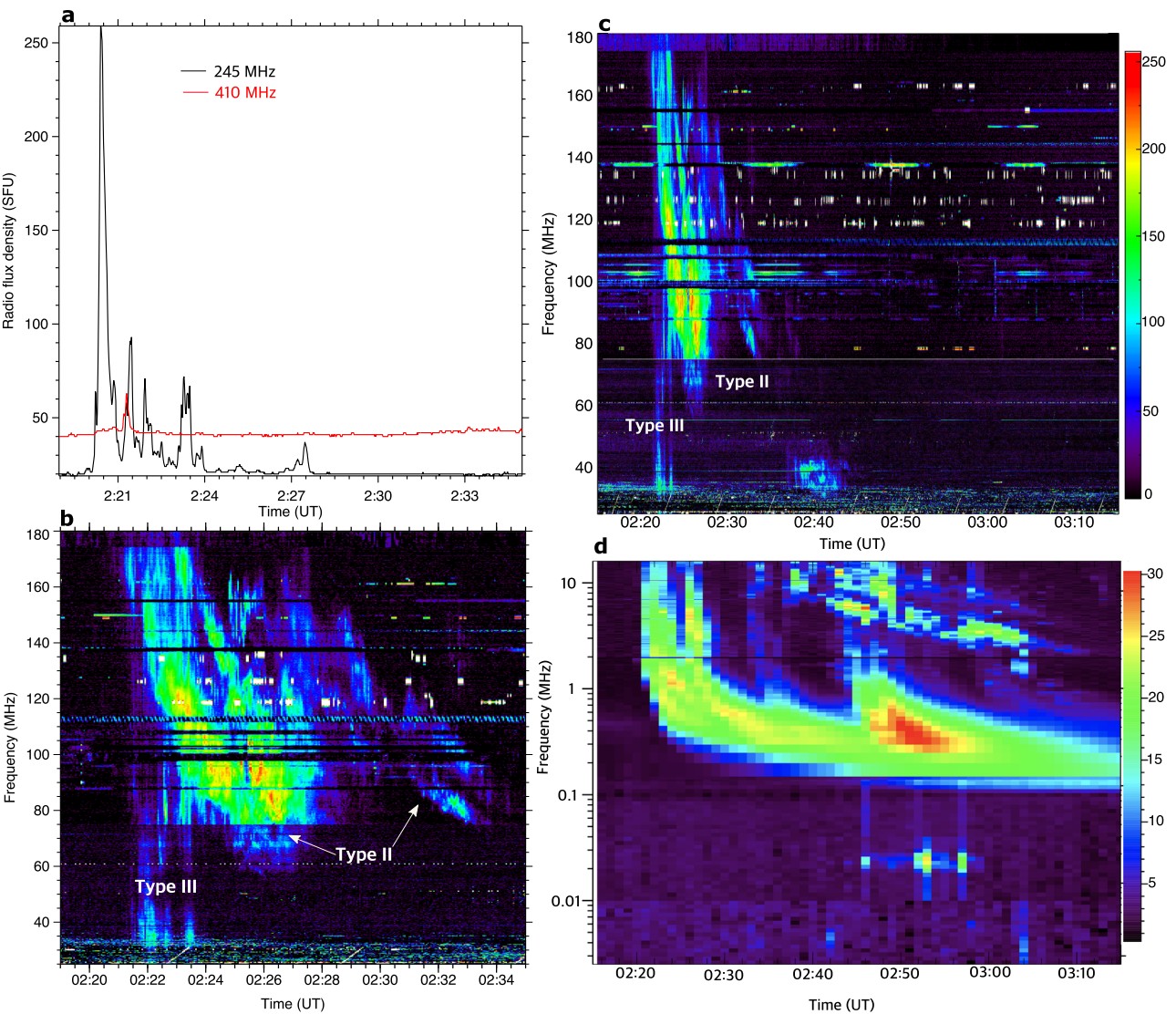

**Fig. 5 | Radio bursts associated with the eruptive flare and the encounter between the EUV wave and the pseudostreamer on May 9, 2014. a** RSTN 1-s cadence radio flux density in 245 (black) and 410 (red) MHz from Learmonth radio observatory. 1 SFU (solar flux unit)=$10^{-22}$ Wm$^{-2}$Hz$^{-1}$. **b–d** Dynamic radio spectra from Learmonth radio observatory (25–180 MHz) and STEREO-A WAVES (2.6 kHz-16.025 MHz). The color bar for (**b**, **c**) is displayed in arbitrary units, whereas the color bar unit for (**d**) represents the average intensity of the electric field (in decibels (dB) above the background level).

limb. The angular separation between SDO and STEREO-B was 165° on May 9, 2014. We utilized EUVI-B 195 Å images, captured at a 5-minute cadence, to infer the magnetic configuration of the eruption site and pseudostreamer from this alternate viewing angle. The size of the STEREO/EUVI image is 2048 × 2048 pixels (1.6" per pixel) encompassing a field of view extending up to 1.7 $R_{\odot}$.

**MHD simulation of mode conversion**

For an illustration of MHD mode conversion in the vicinity of a null point, we consider the interaction of a fast magnetoacoustic wave with a null point given by the magnetic field

$$\vec{B} = \left( -\frac{B_0 x}{L}, 0, \frac{B_0 z}{L} \right), \tag{1}$$

where $B_0$ is a characteristic field strength and $L$ is the length scale for magnetic field variations. $B_0$ and $L$ are constant. In the initial equilibrium, the density and temperature are uniform and there are no steady flows. At a large distance from the null point the plasma $\beta$ is

small, while at the distance of one spatial unit from the origin $\beta$ is unity. The evolution of this equilibrium was numerically modeled by solving the ideal MHD equations with the 2D Lagrangian remap code LARE2D[51]. The boundary condition driving representing the incoming fast wave was implemented by explicitly forcing the MHD parameters to the values expected for an inward propagation of a fast wave, such that

$$\vec{V} = (V_d \cos(\alpha) \sin(\omega t), 0, V_d \sin(\alpha) \sin(\omega t)), \tag{2}$$

where $t$ is time, $\omega$ is the angular frequency, $\alpha$ is the chosen angle of propagation, and $V_d$ is the amplitude. The initial relative amplitude $V_d$ is 0.015, and the oscillation frequency $\omega = 1.26$. The driven boundary was localised in a region around the $x = z$ line with a propagation angle $\alpha = -\pi/2$. This computational setup is similar to one used in[4].

Plots representing the temporal evolution of a typical simulation result are shown in Fig. 7. Different panels correspond to different times (arbitrary units) in the simulation. Panel (a) shows a fast wave (F), launched in the upper right quadrant, approaching the null point situated at the origin. The wrapping of the fast-wave front

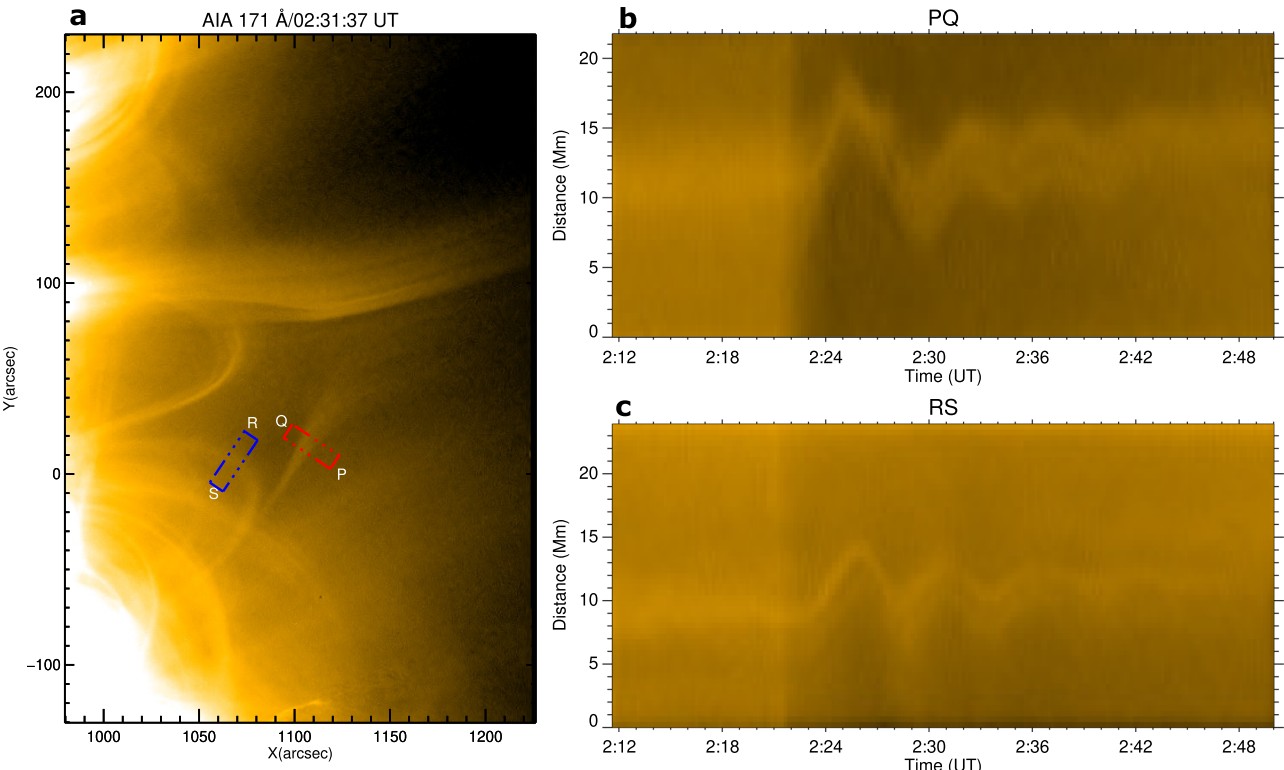

**Fig. 6 | Transverse oscillations of open/closed structures of the pseudostreamer. a–c** Time-distance intensity plots along slices PQ and RS marked in the SDO/AIA 171 Å image. An animation of this Figure is available online as Supplementary Movie 5. The animation runs from 02:12 UT to 02:50 UT.

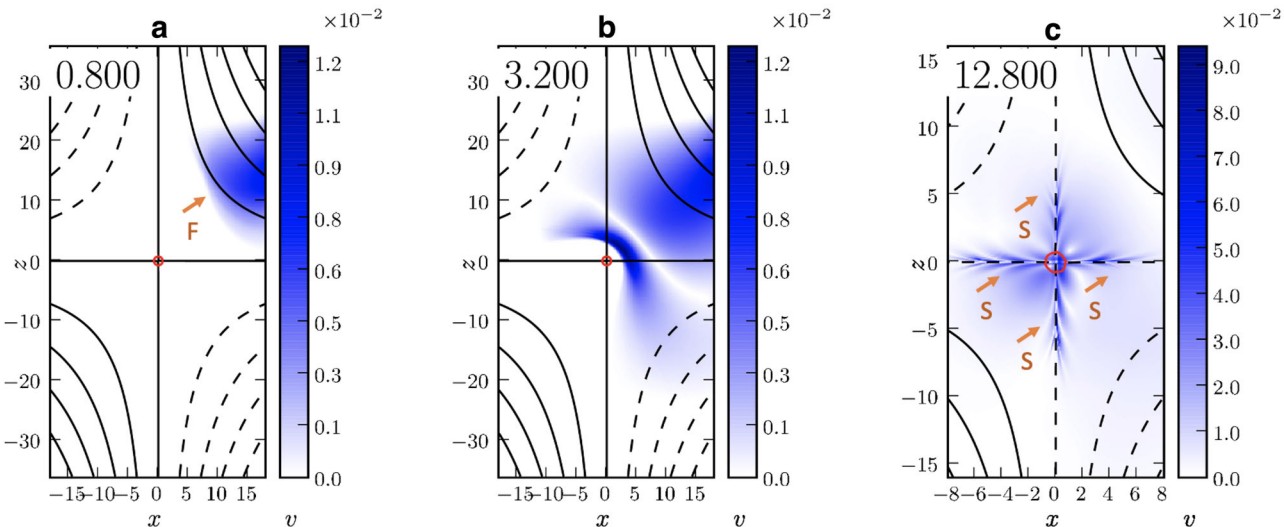

**Fig. 7 | MHD simulation of fast magnetoacoustic-wave mode conversion at a magnetic null point. a–c** Selected panels display filled contour plots representing the absolute velocity values at three different times: during the arrival of the fast-mode wave (F), its wrapping around the null point, and the emergence of slow-mode waves (S, indicated by arrows). The spatial and time units are arbitrary. The black lines show selected magnetic field lines in the plane of the simulation. The solid and dashed black lines indicate the field lines in the opposite quadrants. The red circles show the $\beta = 1$ contour. For this simulation, $\omega = 1.2$.

around the null is evident in panel (b). The slow waves, S, propagating outward from the null point along the separatrices are indicated in panel (c).

## Magnetic Configuration

To determine the magnetic configuration of the source region, we employed a potential-field extrapolation code[52] from the GX simulator package of SSWIDL[53]. The code was applied to a magnetogram obtained by SDO HMI at 01:00:36 UT on May 5, 2014, four days prior to the eruption (Fig. 8a). The potential-field extrapolation of the source region (Fig. 8b) reveals the fan-spine/null-point topology. The peak value of the photospheric magnetic field (positive/negative) in the displayed field of view was ± 600 G. We stress that pseudostreamers typically persist for several days to weeks, as long as the central minority polarity exists. For the event studied here, we tracked the pseudostreamer for 4 days (May 5 to May 9) using AIA 211/171 Å images. The pseudostreamer was present during all 4 days.

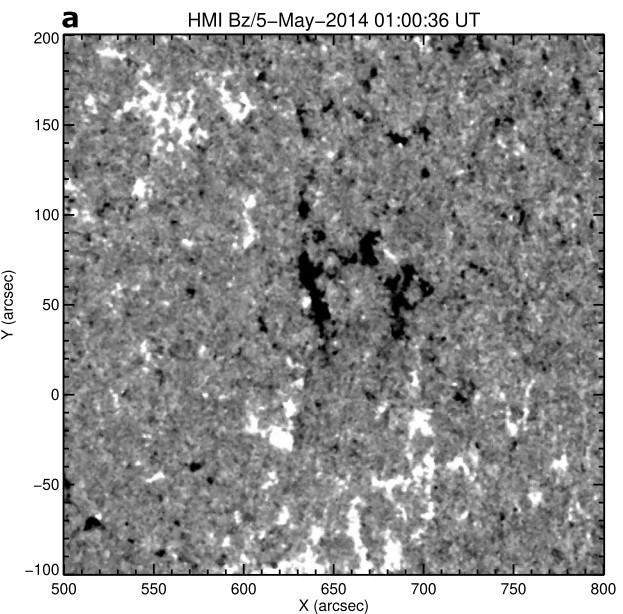
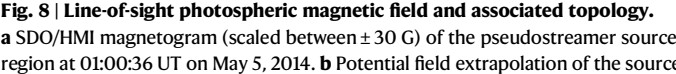
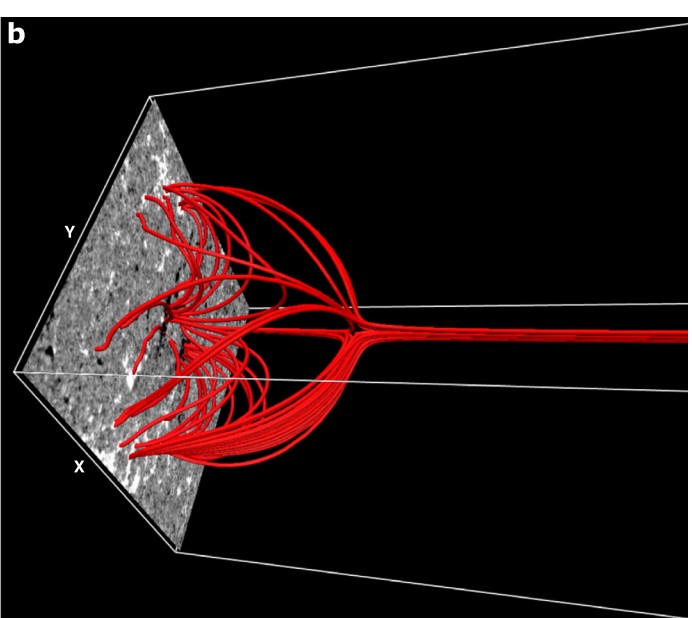

**Fig. 8 | Line-of-sight photospheric magnetic field and associated topology.**
**a** SDO/HMI magnetogram (scaled between ± 30 G) of the pseudostreamer source region at 01:00:36 UT on May 5, 2014. **b** Potential field extrapolation of the source region showing fan-spine (null-point) topology. The field of view (300″ × 300″) is equivalent to that in (**a**).

## Reporting summary

Further information on research design is available in the Nature Portfolio Reporting Summary linked to this article.

## Data availability

The data utilized in this study are publicly available in the Joint Science Operations Center (JSOC) database (http://jsoc.stanford.edu/ajax/exportdata.html?ds=aia.lev1_euv_12s), Virtual Solar Observatory (VSO) database (https://sdac.virtualsolar.org/cgi/search), and NASA Coordinated Data Analysis Web (CDAWeb) (https://cdaweb.gsfc.nasa.gov/). The solar radio burst data is publicly available at NOAA National Geophysical Data Center (https://www.ngdc.noaa.gov/stp/space-weather/solar-data/solar-features/solar-radio/rstn-spectral/). The data that support the findings of this study (or generated during the analysis) are available from the corresponding author upon request.

## Code availability

We analysed data using the Interactive Data Language (IDL), Solar-SoftWare (SSW) package. The routines used to process and analyse data are publicly available in the SSWIDL libraries (https://www.lmsal.com/sdodocs/doc/dcur/SDOD0060.zip/zip/entry/). The LARE2D code used for MHD simulation is publicly available at (https://github.com/Warwick-Plasma/Lare2d). The magnetic-field extrapolation was visualized with VAPOR. VAPOR is a product of the Computational Information Systems Laboratory at the National Center for Atmospheric Research.

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

## Acknowledgements

SDO is a mission for NASA's Living With a Star (LWS) program. The SECCHI instrument was constructed by a consortium of international institutions: the Naval Research Laboratory (USA), the Lockheed Martin Solar and Astrophysical Laboratory (USA), the NASA Goddard Space Flight Center (USA), the Max-Planck Institute for Solar System Research (Germany), the Center Spatial de Liege (Belgium), the University of Birmingham (UK), the Rutherford Appleton Laboratory (UK), the Institut dÓptique (France), and the Institute dÁstrophysique Spatiale (France). This research was supported by NASA's Heliophysics Guest Investigator (#80NSSC20K0265), Supporting Research (#80NSSC24K0264), and GSFC Internal Scientist Funding Model (H-ISFM) programs, and NSF grant #2229336. V.M.N. acknowledges support from the Latvian Council of Science Project No. lzp2022/1-0017.

## Author contributions

P.K. identified the event, analysed the data, produced figures/animations, and wrote the first draft. V.M.N. provided the MHD simulation and contributed to the writing of the paper. J.T.K. contributed to the interpretation and writing of the paper. K.S.C. helped interpret the radio observations. All co-authors contributed to the scientific discussion of the results, and editing/reviewing the manuscript.

## Competing interests

The authors declare no competing interests.
