## [Peer Review File · Nature Communications]

Direct Imaging of MHD Wave Mode Conversion Near a 3D Null Point in the Solar CoronaReviewers' Comments:

Reviewer #1 (Remarks to the Author):

The observations are interesting and the movies with difference images are impressive. We see very well the transverse oscillations produced by the EUV wave of the flare close by. We see effectively different structures which look like in models of reconnection in 2D. According to the shape of the structures visible in the corona, you guess that there is a pseudo-streamer and that the conversion of fast to slow magneto waves could be due to the change of pressure. This kind of event has already been observed in other studies as you quote (Chandra et al, Chen et al). The description of your observations is well done. However the use of some technical words to describe your observations is non sense. Separatrix, null point according to their definition require the knowledge of the magnetic field in the corona using extrapolation; In your case you cannot get magnetograms on the limb of the active region and therefore it is impossible to use these words. At most you may suggest that it can be such magnetic configuration with a null point. So the paper has to be revised completely to be honest and clear with moderate sentences (look-like, conjecture, ..). May be it is not so imperative to be published in Nature communication. It is nice observations like previous publications but nothing really new if we can not accept your study as reality.

Reviewer #2 (Remarks to the Author):

This report is about the manuscript titled 'First Imaging of MHD Wave Mode Conversion Near a 3D Null Point in the Solar Corona'. As described in the title, this study is about the first ever observation for an mhd wave mode conversion at a null point in the solar corona. The authors combined observations by SDO/AIA and STEREO EUVI-B to track a flare generated EUV fast wave propagating through a pseudostreamer, in the presence of a magnetic null point. Running-difference images and animations of the propagating fast wave reveal the generation of slow-mode wavefronts propagating along the magnetic fieldlines, while a kink oscillation was initiated in nearby open and closed magnetic field structures.

The authors did a good job describing the phenomenon and providing clear observations of it. Using Learmonth radio observatory and STEREO A WAVES data of periodic type II and III radio bursts with initiation times matching the perturbation of the null point by the fast wave and the launch of the slow waves, the authors also suggest that the interaction of the fast wave with the null point induced quasi-periodic reconnection, a phenomenon that has been so far been predicted by numerical studies.

The data are well presented and the authors have given logical and plausible interpretations of their

results. To the best of my knowledge, this is the first time wave mode conversion has been directly observed in the solar corona and as such this result is of great importance to the community. I only have a moderate comment and a series of minor comments that I would like the authors to address. I would be happy to reread the updated version of this article before giving my final suggestion.

Main comment

1) Second to last paragraph from the end of page 7: 'indicating a possible connection between the kink oscillation and the onset of the slow mode wave near the null.'

-- I am not entirely certain what the authors meant to say here. Indeed we see a correlation between the kink oscillations and the onset of the slow mode waves at the null. However, there is no proof provided in the manuscript of a cause and effect relation between these two. In my opinion, the fact that the initiation times for the slow EUV waves S1 and S2 match closely with the first two cycles of the oscillations is not enough to prove causality. The two effects could be either independent phenomena caused by the original EUV fast wave front (e.g. Nakariakov et al. 1999, *Science*, 285, 862; McLaughlin et al. 2009, *A&A*, 493, 227), or one could be "driving" the other (as in the model by Nakariakov et al. 2006, *A&A* 452, 343). I agree with the authors that this goes beyond the scope of this study and should be investigated further for a proper conclusion to be reached. Therefore, I would argue that this needs to be at least worded differently.

Minor comments

1) The movie 454096_0_video_89874_rx35nk.mp4 shows the decaying kink oscillations along the slices PQ and RS, as mentioned in the beginning of the last paragraph of Section 2. This movie is not directly associated with any figure in the manuscript. I would suggest connecting this movie to Figure 1, to add the slices PQ and RS on Figure (1b) and also add the two panels for the time-distance maps of the kink oscillations on Figure 1.

2) In Figures 1b, 3a, and 4d we see the approximate location of the null point being highlighted. However, there is no mention in the text how this location was identified. Was this done purely by visual inspection?

3) Appendix A., page 18: 'The driven boundary was localised in a region around the $x = z$ line with a propagation angle $\alpha = -\pi/2$. This computational setup is similar to one used in Nakariakov et al. [23].'

-- Although this is outside the main scope of the paper, it would be interesting to the readers to have some additional information on the numerical setup presented here. For example, how wide was the localised driving region around the $x=z$ line. Also, wouldn't $\alpha = -\pi/2$ suggest that the excited fast wave is only directed along the $-z$ direction instead of towards the null point?

4) Last paragraph of Section 2: 'The initial amplitude and oscillation period of the oscillation...' -> 'The initial amplitude and period of the oscillation...'

5) Figure A.1., caption: 'Slow-mode waves are labeled by red arrows and S1-S3.'

-- In panel c, we can identify the slow-mode waves along the separatrices, but there is no individual labeling. Perhaps the authors meant to draw an analogy between these and the S1, S2 and S3 from the

observations? I suggest rephrasing this last sentence.

Reviewer #3 (Remarks to the Author):

This paper presents an observation of MHD wave mode conversion in the vicinity of a magnetic null point in the solar corona. While this phenomenon has been the subject of theoretical studies for many years, this marks the first time it has been directly observed in the corona. This is significant not only because it lends empirical support to the existing theoretical investigations of wave activity near null points, but it also provides concrete evidence that the mode conversion mechanism operates within the solar atmosphere. This process is also believed to be in effect in the lower atmosphere at the $\beta=1$ equipartition layer. Given its relevance from both observational and theoretical perspectives, I believe that this paper is well-suited for publication in Nature Communications.

Nonetheless, I do have a few comments, and I kindly request the authors to address them in their revision. None of these issues present unbeatable obstacles and, probably, they would not prevent me from ultimately recommending the paper for publication. Nevertheless, I encourage the authors to consider these points carefully in their revision. They are given in no particular order.

(1) The observations are meticulously detailed and presented in a convincing manner. However, I believe there is room for further development in the theoretical explanation of the observed process and its potential implications. One aspect that raises my interest is that two upward-moving slow mode fronts (S1 and S2) are observed but a single downward-moving front (S3) is reported. It may suggest that the mode conversion process appears to favor the generation of upward-propagating slow waves. Could the pseudostreamer magnetic topology or the angle of attack of the fast wave influence this phenomenon? I would appreciate it if the authors could provide some insights or discuss possible reasons for this observed asymmetry.

(2) The authors have included an illustrative MHD simulation in the appendix to demonstrate mode conversion in an idealized scenario featuring a null point. I acknowledge their efforts in providing this theoretical model. However, it is worth noting that the configuration explored in the appendix significantly differs from the observations presented in the main body of the paper. As a result, performing a detailed comparison between the observations and simulations is not possible. While the authors do not aim for an in-depth comparison, which would have been a great addition to the paper to be honest, it is important nevertheless to highlight the limitations of the simulation provided in the appendix. For example, the magnetic configuration in the numerical model, while incorporating a null point, substantially deviates from the observed pseudostreamer. Additionally, the numerical simulation does not account for effects such as plasma stratification and gravity. To provide a better theoretical context for their observations, the authors could benefit from referencing prior studies. Some works, such as those by Santamaria et al. (2015, A&A 577), Tarr et al. (2017, ApJ 837), Tarr & Linton (2019, ApJ 879), and Yadav et al. (2022, A&A 660), employ more realistic configurations and could offer valuable insights to complement the theoretical explanations.

(3) Besides the mode conversion observation, the authors discuss the appearance of a global oscillation of the pseudostreamer structure following the passage of the fast wave. However, it remains unclear whether this oscillation is intrinsically linked to the generation of the mode-converted slow waves, as implied by the authors. While it is evident that both phenomena originate from the incident fast wave, establishing a definitive connection between them requires further investigation. In light of this, I would advise the authors to exercise caution when addressing this potential link and explicitly acknowledge in the paper that, at this stage, any such connection remains speculative.

(4) The authors employ the term "kink oscillation" to describe the global oscillation mentioned in the previous comment. While I understand the authors' intent in using this term, it can be considered a misnomer. In the context of coronal oscillations, a "kink oscillation" typically refers to the transverse oscillation of a magnetic flux tube, involving a lateral displacement of the tube axis. The term "kink" specifically corresponds to the azimuthal wavenumber $m=1$ in a cylindrical coordinate system aligned with the tube axis. Since the pseudostreamer structure significantly differs from the standard coronal flux tube model, I would recommend that the authors opt for a more generic term, such as "global oscillation," "transverse oscillation," or "lateral oscillation," to avoid any geometrical or theoretical implications associated with the term "kink" that might potentially mislead readers.

Dear reviewers,

We thank for the constructive comments/criticisms on our paper that have guided our revisions. We respectfully disagree with the majority of the comments from Reviewer #1 and offer compelling responses, supported by evidence, for each of the comments below. The revised text in the manuscript is in bold fonts. We added new Figures (Fig. 6 & Appendix Fig. 1B) to clarify the transverse oscillations and magnetic topology of the studied pseudostreamer.

Reply to reviewer#1 comments:

The observations are interesting and the movies with difference images are impressive. We see very well the transverse oscillations produced by the EUV wave of the flare close by. We see effectively different structures which look like in models of reconnection in 2D. According to the shape of the structures visible in the corona, you guess that there is a pseudo-streamer and that the conversion of fast to slow magneto waves could be due to the change of pressure. This kind of event has already been observed in other studies as you quote (Chandra et al, Chen et al). The description of your observations is well done.

Reply: Thank you for the constructive comments/criticism. In the introduction, we explicitly highlighted the distinctions between our observations and previous studies. Specifically, neither Chandra et al. nor Chen et al. provided direct imaging of mode conversion at a 3D null point. Their inferences of the mode conversion (mostly indirect) process were based on the stopping of fast EUV wavefronts at the separatrix (open-closed boundary) in different structures (not at a 3D null).

To the best of our knowledge, nobody has previously presented observational evidence of mode conversion at a 3D null, specifically the presence of multiple propagating slow-mode EUV waves along the pseudostreamer stalk after the passage of a fast-mode wave. We present the first direct imaging of the mode conversion process at a 3D null, and the other two reviewers concur with our findings.

However the use of some technical words to describe your observations is non sense. Separatrix, null point according to their definition require the knowledge of the magnetic field in the corona using extrapolation; In your case you cannot get magnetograms on the limb of the active region and therefore it is impossible to use these words. At most you may suggest that it can be such magnetic configuration with a null point.

At most you may suggest that it can be such magnetic configuration with a null point. So the paper has to be revised completely to be honest and clear with moderate sentences (look-like, conjecture, ..).

Reply: There is a wealth of literature showing that our use of such topological terminology is correct. The referee is correct that we are inferring the presence of a null, with associated separatrix, but this is a reasonable conjecture based on our previous studies of fan-spine topologies (including pseudostreamers) on-disk and at the limb (e.g., Kumar et al. 2021). These null-point topologies are prevalent in the solar corona and have been extensively studied by the solar research community (e.g., Masson et al. 2014, Karna et al. 2019, Mason et al. 2019). Therefore, we are not employing any unusual or unknown terminology.

To address this criticism, we have now included a potential field extrapolation (New Figure B.1 in the appendix) to illustrate that this configuration is a pseudostreamer with a null. We utilized an HMI magnetogram taken 4 days prior to the eruption when the source region was on the solar disk. It is important to note that the fundamental magnetic structure of pseudostreamers and their appearance in EUV images are well-established in the solar community.

Maybe it is not so imperative to be published in Nature communication. It is nice observations like previous publications but nothing really new if we can not accept your study as reality.

We hope that our revision has addressed the concerns raised by the reviewer, and we are open to considering any additional suggestions for improvement.

Reviewer #2 (Remarks to the Author):

This report is about the manuscript titled 'First Imaging of MHD Wave Mode Conversion Near a 3D Null Point in the Solar Corona'. As described in the title, this study is about the first ever observation of an MHD wave mode conversion at a null point in the solar corona. The authors combined observations by SDO/AIA and STEREO EUVI-B to track a flare generated EUV fast wave propagating through a pseudostreamer, in the presence of a magnetic null point. Running-difference images and animations of the propagating fast wave reveal the generation of slow-mode wavefronts propagating along the magnetic field lines, while a kink oscillation was initiated in nearby open and closed magnetic field structures.

The authors did a good job describing the phenomenon and providing clear observations of it. Using Learmonth radio observatory and STEREO A WAVES data of periodic type II and III radio bursts with initiation times matching the perturbation of the null point by the fast wave and the launch of the slow waves, the authors also suggest that the interaction of the fast wave with the null point induced quasi-periodic reconnection, a phenomenon that has been so far been predicted by numerical studies.

The data are well presented and the authors have given logical and plausible interpretations of their results. To the best of my knowledge, this is the first time wave mode conversion has been directly observed in the solar corona and as such this result is of great importance to the community.

I only have a moderate comment and a series of minor comments that I would like the authors to address. I would be happy to reread the updated version of this article before giving my final suggestion.

Reply: Thank you for your constructive comments and suggestions.

Main comment

1) Second to last paragraph from the end of page 7: 'indicating a possible connection between the kink oscillation and the onset of the slow mode wave near the null.'

-- I am not entirely certain what the authors meant to say here. Indeed we see a correlation between the kink oscillations and the onset of the slow mode waves at the null. However, there is no proof provided in the manuscript of a cause and effect relation between these two. In my opinion, the fact that the initiation times for the slow EUV waves S1 and S2 match closely with the first two cycles of the oscillations is not enough to prove causality. The two effects could be either independent phenomena caused by the original EUV fast wave front (e.g. Nakariakov et al. 1999, Science, 285, 862; McLaughlin et al. 2009, A&A, 493, 227), or one could be "driving" the other (as in the model by Nakariakov et al. 2006, A&A 452, 343). I agree with the authors that this goes beyond the scope of this study and should be investigated further for a proper conclusion to be reached. Therefore, I would argue that this needs to be at least worded differently.

Reply: We agree that the transverse oscillation and the appearance of slow-mode waves are simultaneous. The connection between the two is speculative. Both might be independent phenomena triggered by the incoming fast EUV wave. We have revised the sentence accordingly. Future 3D MHD simulations should be conducted to understand a clear connection between these two phenomena.

Minor comments

1) The movie 454096_0_video_89874_rx35nk.mp4 shows the decaying kink oscillations along the slices PQ and RS, as mentioned in the beginning of the last paragraph of Section 2. This movie is not directly associated with any figure in the manuscript. I would suggest connecting this movie to Figure 1, to add the slices PQ and RS on Figure (1b) and also add the two panels for the time-distance maps of the kink oscillations on Figure 1.

Reply: A very good point. We agree. We included a separate figure (New Figure 6) showing transverse oscillations TD plots.

2) In Figures 1b, 3a, and 4d we see the approximate location of the null point being highlighted. However, there is no mention in the text how this location was identified. Was this done purely by visual inspection?

Reply: This is an approximate location of the null, done by visual inspection. To strengthen this identification we have performed a potential field extrapolation that shows a clear null-point topology and added it to the manuscript (New Figure B.1).

3) Appendix A., page 18: 'The driven boundary was localised in a region around the $x = z$ line with a propagation angle $\alpha = -\pi/2$. This computational setup is similar to one used in Nakariakov et al. [23].'

-- Although this is outside the main scope of the paper, it would be interesting to the readers to have some additional information on the numerical setup presented here. For example, how wide was the localised driving region around the $x=z$ line. Also, wouldn't $\alpha = -\pi/2$ suggest that the excited fast wave is only directed along the $-z$ direction instead of towards the null point?

Reply: As it was demonstrated in the series of papers of McLaughlin and co-authors, the actual direction of the fast wave vector at a distance from the null point is not important. The fast wave front "wraps around" the X-point because of the refraction, resulting in the propagation towards the null point.

4) Last paragraph of Section 2: 'The initial amplitude and oscillation period of the oscillation...' -> 'The initial amplitude and period of the oscillation...'

Reply: We corrected the sentence.

5) Figure A.1., caption: 'Slow-mode waves are labeled by red arrows and S1-S3.'

-- In panel c, we can identify the slow-mode waves along the separatrices, but there is no individual labeling. Perhaps the authors meant to draw an analogy between these and the S1, S2 and S3 from the observations? I suggest rephrasing this last sentence.

Reply: We revised the sentence.

Reviewer #3 (Remarks to the Author):

This paper presents an observation of MHD wave mode conversion in the vicinity of a magnetic null point in the solar corona. While this phenomenon has been the subject of theoretical studies for many years, this marks the first time it has been directly observed in the corona. This is significant not only because it lends empirical support to the existing theoretical investigations of wave activity near null points, but it also provides concrete evidence that the mode conversion mechanism operates within the solar atmosphere. This process is also believed to be in effect in the lower atmosphere at the $\beta=1$ equipartition layer. Given its relevance from both observational and theoretical perspectives, I believe that this paper is well-suited for publication in Nature Communications.

Nonetheless, I do have a few comments, and I kindly request the authors to address them in their revision. None of these issues present unbeatable obstacles and, probably, they would not prevent me from ultimately recommending the paper for publication. Nevertheless, I encourage the authors to consider these points carefully in their revision. They are given in no particular order.

Reply: Thank you for your constructive comments/suggestions.

(1) The observations are meticulously detailed and presented in a convincing manner. However, I believe there is room for further development in the theoretical explanation of the observed process and its potential implications. One aspect that raises my interest is that two upward-moving slow mode fronts (S1 and S2) are observed but a single downward-moving front (S3) is reported. It may suggest that the mode conversion process appears to favor the generation of upward-propagating slow waves. Could the pseudostreamer magnetic topology or the angle of attack of the fast wave influence this phenomenon? I would appreciate it if the authors could provide some insights or discuss possible reasons for this observed asymmetry.

Reply: These are excellent points. Please note that the upward-moving front S2 is very faint and disappears shortly after propagating ~30 Mm. There may be a downward-propagating counterpart, but it may not have been detected due to the weak signal or limited resolution (i.e., fainter front propagating downward through the denser fan loops).

In principle, as indicated by MHD simulations (please see reply #2 for more detail), multiple slow-mode fronts simultaneously propagate upward and downward along the separatrix. We generally observe a similar mode conversion process in a limited region around the null (as shown in the paper) or within the entire pseudostreamer (e.g., Tarr & Linton 2019). Therefore, it is unlikely that the magnetic topology or angle of attack will significantly affect the mode conversion phenomena.

(2) The authors have included an illustrative MHD simulation in the appendix to demonstrate mode conversion in an idealized scenario featuring a null point. I acknowledge their efforts in providing this theoretical model. However, it is worth noting that the configuration explored in the appendix significantly differs from the observations presented in the main body of the paper. As a result, performing a detailed

comparison between the observations and simulations is not possible. While the authors do not aim for an in-depth comparison, which would have been a great addition to the paper to be honest, it is important nevertheless to highlight the limitations of the simulation provided in the appendix. For example, the magnetic configuration in the numerical model, while incorporating a null point, substantially deviates from the observed pseudostreamer. Additionally, the numerical simulation does not account for effects such as plasma stratification and gravity. To provide a better theoretical context for their observations, the authors could benefit from referencing prior studies. Some works, such as those by Santamaria et al. (2015, A&A 577), Tarr et al. (2017, ApJ 837), Tarr & Linton (2019, ApJ 879), and Yadav et al. (2022, A&A 660), employ more realistic configurations and could offer valuable insights to complement the theoretical explanations.

Reply: Thank you for the valuable references. We agree that our 2D MHD simulation, presented here, concentrates on the behavior of the fast-mode wave in a limited region around the null. However, it's worth noting that other MHD simulations cited in the paper have explored the interaction of the fast-mode wave through the pseudostreamer null-point and found nearly consistent results.

Our numerical 2D MHD simulation (velocity component is shown) demonstrates the general behavior of a fast-mode wave at the null point. The incoming fast-mode wavefront undergoes complex deformation due to the complex magnetic topology near the null point. Slow-mode wavefronts appear after the interaction of the fast-mode wave at the null point. The slow-mode wavefronts propagate outwards from the null along the separatrices. The observations are essentially consistent with the MHD simulation.

Other MHD simulations offer evidence of mode conversion of MHD waves in PS null-point topologies (Santamaria et al. 2017, Tarr et al. 2017, Tarr & Linton 2019, Yadav et al. 2022). For instance, Tarr & Linton (2019) showed a clear interaction of fast mode waves through a PS null in a 2D MHD model. They detected slow-mode waves propagating upward along the PS stalk and simultaneously downward along the separatrix after the interaction (animation), similar to the observations reported here. However, they found no evidence of oscillatory reconnection arising from the dynamics of the null itself. In addition, the above simulations do not provide any evidence for transverse oscillations of the PS open/closed structures. More realistic future 3D MHD simulations may contribute to a better understanding of the interaction of MHD waves through null-point topologies and the associated quasiperiodic/oscillatory reconnection in greater detail. It is important to note that the interaction of MHD waves through the null leads to mode conversion, followed by oscillatory reconnection. This insight may aid in understanding quasiperiodic pulsations, particle acceleration, and recurrent jet outflows from the null.

We included the above discussion in the revised version.

(3) Besides the mode conversion observation, the authors discuss the appearance of a global oscillation of the pseudostreamer structure following the passage of the fast wave. However, it remains unclear whether this oscillation is intrinsically linked to the generation of the mode-converted slow waves, as implied by the authors. While it is evident that both phenomena originate from the incident fast wave, establishing a definitive connection between them requires further investigation. In light of this, I would advise the authors

to exercise caution when addressing this potential link and explicitly acknowledge in the paper that, at this stage, any such connection remains speculative.

Reply: This is a valid point. We agree that the link between transverse oscillations and slow mode waves is based on speculation. This aspect requires further exploration through 3D MHD simulations. We have made the necessary revisions to the text.

(4) The authors employ the term "kink oscillation" to describe the global oscillation mentioned in the previous comment. While I understand the authors' intent in using this term, it can be considered a misnomer. In the context of coronal oscillations, a "kink oscillation" typically refers to the transverse oscillation of a magnetic flux tube, involving a lateral displacement of the tube axis. The term "kink" specifically corresponds to the azimuthal wavenumber $m=1$ in a cylindrical coordinate system aligned with the tube axis. Since the pseudostreamer structure significantly differs from the standard coronal flux tube model, I would recommend that the authors opt for a more generic term, such as "global oscillation," "transverse oscillation," or "lateral oscillation," to avoid any geometrical or theoretical implications associated with the term "kink" that might potentially mislead readers.

Reply: We replaced "kink oscillation" by "transverse oscillation". However, we would like to point out that the term "kink oscillation" has been used for transverse oscillations of a plasma slab in the seminal paper of Edwin & Roberts. The term "transverse oscillation" may be confusing too, as, for example, sausage oscillations are transverse too. We clarified it in the manuscript.

REVIEWER COMMENTS

Reviewer #1 (Remarks to the Author):

My main comment was on the use of some sentences that were not justified. Now the main body of the paper has been well changed concerning the wave transformation but again they neglect to quote recent papers (see below).
List of Comments:

1. They get around the problem of using the term of null point just looking at observations. Now they show that it is possible that there exists a null point using a potential extrapolation. They should mention if the existence of null point is stable during 4 days before the observations in this region with a so weak field (30 Gauss). For that they need to give the common time of network evolution.
2. The second comment is about the abstract. They have not adapted the abstract to the new text. The following sentence is really a non justified statement :
"These novel observations confirm the existence of mode conversion near 3D nulls in the solar corona,"
It is now well smoothed in the text that the transverse waves are concomitant with the fast wave but the transformation is very speculative as it is written in the following sections:
Section 2 : "A transverse oscillation (i.e., a kink or anti-symmetric transverse oscillation, appeared shortly after the passage of the fast EUV wave through the PS."
Conclusion : "The connection between transverse oscillation and the appearance of slow mode waves is speculative here."
The link between the transformation of fast wave and the generation of the mode-converted slow waves is not clear.
In the abstract they should change the words "confirm" by "suggest "(or "are consistent") that it could be similar to simulations.
3. Concerning the simulations, it is written in their paper: More realistic simulations are needed to confirm the interaction of fast waves with null point topologies; Such simulations exist.
In the paper of Sabri et al 2023 (and others referenced herein) on "Propagation of the Alfvén Wave and Induced Perturbations in the Vicinity of a 3D Proper Magnetic Null Point" it is explained that fast magneto acoustic wave refracts at the null point in the fan plane. Also, the slow waves accumulate and both participate to the heating. 3D simulations is very different from 2D.
They should discuss these papers.
4. General comment
Such strong statements (Title, abstract) are definitively not relevant in a research paper for the Nature Journal.
I still disagree with such strong affirmations.
The title is misleading. Because it does not reflect what is saying in the paper which is now better written with more nuances and more speculations. The simulation itself does not reflect the observations as they mention.

The paper could be acceptable with a new title and an abstract reflecting the main body of the paper

which should stay as an observational detailed paper. Their simulations are still too simple.

Reviewer #2 (Remarks to the Author):

This report is about the revised manuscript titled 'First Imaging of MHDWave Mode Conversion Near a 3D Null Point in the Solar Corona', reporting the first ever observation of fast to slow mhd wave mode conversion at a null point in the solar corona, the propagation of slow-mode wavefronts along the magnetic fieldlines of a pseudostreamer, the manifestation of kink oscillations in the open and closed magnetic field structures and finally, evidence suggesting the presence of periodic magnetic reconnection.

To the best of my knowledge, the results are novel and the process described has not been reported in past observations. The authors have also done a good job addressing the comments raised. The addition of Appendix B in particular verifies very effectively the previously assumed magnetic field configuration, strengthening the interpretation presented by the authors. I only have a minor comment that I am stating below. Apart from that, I am happy to recommend this work for publication in Nature Communications, as long as my comment has been addressed.

Minor comment:

Second to last paragraph in section 3:

"We find that the transformation of MHD waves at nulls leads to mode conversion followed by oscillatory reconnection."

-- This needs to be rephrased. The simulation included in Appendix A does not explicitly show the manifestation of periodic reconnection, but only the effects of mode conversion and the slow wavefront propagation along the separatrices. However, mode conversion alongside oscillatory reconnection at null points has been shown in the past in other works (see Thurgood et al. 2017 [38] for 3D null points). I would suggest the authors make a few additions to that part of the text.

Reviewer #3 (Remarks to the Author):

I thank the authors for taking my comments into account and for revising the paper accordingly. My concerns have been satisfactorily addressed and I believe that the paper has been improved. I have no further comments. Now, I am happy to recommend the paper for publication.

Dear reviewers,

Thank you for the constructive comments/suggestions on our paper that have guided our revisions. The revised text in the manuscript is in bold fonts.

Thank you

Kind regards

Pankaj Kumar & coauthors

Reviewer #1 (Remarks to the Author):

My main comment was on the use of some sentences that were not justified. Now the main body of the paper has been well changed concerning the wave transformation but again they neglect to quote recent papers (see below).

List of Comments:

1. They get around the problem of using the term of null point just looking at observations. Now they show that it is possible that there exists a null point using a potential extrapolation. They should mention if the existence of null point is stable during 4 days before the observations in this region with a so weak field (30 Gauss). For that they need to give the common time of network evolution.

Reply: Please note that the scale for displaying the B component is set at +/-30 G to visualize weak fields up to +/-30 G. However, the peak value of the photospheric magnetic field (positive/negative) was +/-600 G within the displayed field of view (Figure B1 (a)). It's important to highlight that pseudostreamers persist for several days to weeks, as long as the central minority polarity exists. In our earlier observations (Kumar et al. 2021), we tracked three pseudostreamers in AIA (via Helioviewer) as they traversed from the east limb to the west limb, exhibiting survival for two weeks and even longer.

For the event studied here, we tracked the pseudostreamer for 4 days (May 5 to May 9) using AIA 211/171 images. The pseudostreamer was present during these 4 days. We revised the text in the appendix for clarity.

2. The second comment is about the abstract. They have not adapted the abstract to the new text.

The following sentence is really a non justified statement :

“These novel observations confirm the existence of mode conversion near 3D nulls in the solar corona,”

It is now well smoothed in the text that the transverse waves are concomitant with the fast wave but the transformation is very speculative as it is written in the following sections:

Section 2 : “A transverse oscillation (i.e., a kink or anti-symmetric transverse oscillation, appeared shortly after the passage of the fast EUV wave through the PS.”

Conclusion :”The connection between transverse oscillation and the appearance of slow mode waves is speculative here.”

The link between the transformation of fast wave and the generation of the mode-converted slow waves is not clear.

In the abstract they should change the words “confirm” by “suggest” (or “are consistent”) that it could be similar to simulations.

Reply: We revised the abstract and replaced “confirm” by “suggest”. We did not include this statement in the paper, “*The link between the transformation of fast wave and the generation of the mode-converted slow waves is not clear.*” The mode conversion is clearly detected in the observation as predicted by MHD simulations; there is no speculation. The observations presented in the paper clearly demonstrate the incoming fast EUV wave passing through the null and appearance of slow-mode

waves emerging from the null. The observation of mode conversion is consistent with what we see in the MHD simulations presented in the paper, as well as theoretical modeling performed by other colleagues cited in the discussion. Our observations also reveal additional transverse oscillations in the open/closed structure of the pseudostreamer. It's important to note that the sentences in Section 2 and the conclusions primarily focus on the link between transverse oscillation and the possibility of oscillatory reconnection, not on mode conversion. In the revised abstract, we have included additional sentences addressing transverse oscillations.

3. Concerning the simulations, it is written in their paper: More realistic simulations are needed to confirm the interaction of fast waves with null point topologies; Such simulations exist.

In the paper of Sabri et al 2023 (and others referenced herein) on “Propagation of the Alfvén Wave and Induced Perturbations in the Vicinity of a 3D Proper Magnetic Null Point” it is explained that fast magneto acoustic wave refracts at the null point in the fan plane. Also, the slow waves accumulate and both participate to the heating. 3D simulations is very different from 2D.

They should discuss these papers.

Reply: We believe that the reviewer meant the paper of Sabri et al. 2022ApJ...924..126S. We included Sabri et al. (2022) and other relevant papers in the discussion as per the referee’s suggestion.

The effect of the refraction of fast magnetoacoustic waves at a null point had been theoretically studied in several papers before Sabri et al. (2022). Certainly, the physics appearing in 3D models is richer than in 2D. However, in the context of our paper, results of 2D simulations of the discussed phenomenon are not “very different” from 3D simulations. Specifically, the fast-to-slow mode conversion process occurs in both 2D or 2.5D simulations, as well as in 3D simulations. The main novel outcome of Sabri et al. (2022) is that in 3D the transformation of a fast wave into a slow wave results in a slow wave with a higher amplitude than in 2D. This possible increase in the slow wave amplitude is not relevant to our study, which reports qualitative effects and does not contain energy estimations. The simulation shown in our paper and other simulations cited in the discussion are sufficient to show the interaction of a fast-mode wave with the null and generation of slow-mode waves as a result of the mode conversion process. However, following the suggestion, we now revised the sentence (“*More realistic 3D MHD simulations would contribute to a better understanding of the interaction between MHD waves and null-point topologies, transverse oscillations, and the associated repetitive reconnection.*”) which is more intended to understand the link between transverse oscillation excited by the passage of fast-mode wave and associated oscillatory reconnection via new MHD simulations.

4. General comment

Such strong statements (Title, abstract) are definitively not relevant in a research paper for the Nature Journal. I still disagree with such strong affirmations.

The title is misleading. Because it does not reflect what is saying in the paper which is now better written with more nuances and more speculations. The simulation itself does not reflect the observations as they mention. The paper could be acceptable with a new title and an abstract reflecting the main body of the paper which should stay as an observational detailed paper. Their simulations are still too simple.

Reply: We hold a different opinion on this matter. From our perspective, the title precisely reflects our observational finding, even though we have slightly revised it as per the referee's suggestion. The abstract is clear and concise, explicitly presenting the observational findings and their significance. In response to the reviewer's suggestions, we have included additional sentences in the abstract and replaced ‘confirm’ with ‘suggest’. While the simulations are basic, they effectively illustrate the mode conversion process at the null point in a limited field of view, involving the incoming fast wave, its refraction, and the generation of slow-mode waves. We have compared observations with other simulations in the discussion. The observations will inspire new MHD simulations to understand the

link between transverse oscillations of the pseudostreamer structures and associated oscillatory reconnection. We believe our revisions are sufficient to convince the readers.

Reviewer #2 (Remarks to the Author):

This report is about the revised manuscript titled 'First Imaging of MHD Wave Mode Conversion Near a 3D Null Point in the Solar Corona', reporting the first ever observation of fast to slow mhd wave mode conversion at a null point in the solar corona, the propagation of slow-mode wavefronts along the magnetic fieldlines of a pseudostreamer, the manifestation of kink oscillations in the open and closed magnetic field structures and finally, evidence suggesting the presence of periodic magnetic reconnection.

To the best of my knowledge, the results are novel and the process described has not been reported in past observations. The authors have also done a good job addressing the comments raised. The addition of Appendix B in particular verifies very effectively the previously assumed magnetic field configuration, strengthening the interpretation presented by the authors. I only have a minor comment that I am stating below. Apart from that, I am happy to recommend this work for publication in Nature Communications, as long as my comment has been addressed.

Minor comment:

Second to last paragraph in section 3:

"We find that the transformation of MHD waves at nulls leads to mode conversion followed by oscillatory reconnection."

-- This needs to be rephrased. The simulation included in Appendix A does not explicitly show the manifestation of periodic reconnection, but only the effects of mode conversion and the slow wavefront propagation along the separatrices. However, mode conversion alongside oscillatory reconnection at null points has been shown in the past in other works (see Thurgood et al. 2017 [38] for 3D null points). I would suggest the authors make a few additions to that part of the text.

Reply: We agree. The original sentence was mainly intended for observations. Now we have revised the sentence, and mentioned the Thurgood et al. (2017) simulation paper in the discussion.

Reviewer #3 (Remarks to the Author):

I thank the authors for taking my comments into account and for revising the paper accordingly. My concerns have been satisfactorily addressed and I believe that the paper has been improved. I have no further comments. Now, I am happy to recommend the paper for publication.

Thank you for the recommendation.

REVIEWERS' COMMENTS

Reviewer #2 (Remarks to the Author):

I would like to thank the authors for taking into account and addressing all of my comments during the revision of their submitted paper. I am happy with the current version of the manuscript, and as such, I can recommend it for publication.